# The Reasons for Unfinished Nursing Care during the COVID-19 Pandemic: An Integrative Review

Luisa Sist [1,2], Stefania Chiappinotto [3,*], Rossella Messina [1], Paola Rucci [1] and Alvisa Palese [3]

1   Department of Biomedical and Neuromotor Sciences, Alma Mater Studiorum University of Bologna, 40123 Bologna, Italy; luisa.sist@unibo.it (L.S.); rossella.messina3@unibo.it (R.M.); paola.rucci2@unibo.it (P.R.)
2   IRCCS Azienda Ospedaliero, Universitaria di Bologna, 40138 Bologna, Italy
3   Department of Medicine, University of Udine, 33100 Udine, Italy; alvisa.palese@uniud.it
*   Correspondence: stefania.chiappinotto@uniud.it; Tel.: +39-(0)432-590911

**Abstract: Background:** The concept of unfinished nursing care (UNC) describes nursing interventions required by patients and families that nurses postpone or omit. UNC reasons have been documented; however, no studies have summarised the underlying factors triggering the UNC during the pandemic. Therefore, the aim was to synthesise the available studies exploring factors affecting UNC during a pandemic. **Methods:** We conducted an integrative review following Whittemore and Knafl's framework according to the Reporting Items for Systematic reviews and Meta-Analyses (PRISMA) guidelines. PubMed, the Cumulative Index to Nursing and Allied Health Literature (CINAHL) and the Scopus databases were searched for primary studies that collected data from 1 January 2020 to 1 May 2023. Both qualitative and quantitative studies assessing the reasons for UNC were eligible and evaluated in their quality using the Critical Appraisal Skills Programme and the Mixed Methods Appraisal Tool. **Results:** Four studies were included—three qualitative and one cross-sectional. The reasons for UNC have been documented at the following levels: (a) system (e.g., new healthcare system priorities); (b) unit (e.g., ineffective work processes); (c) nurse management (e.g., inadequate nurse manager's leadership); (d) nurse (e.g., nurses' attitudes, competences, performances); and (e) patient (increased demand for care). **Conclusion:** The reasons for UNC during the COVID-19 pandemic are different to those documented in the pre-pandemic times and reflect a pre-existing frailty of the National Health Service towards nursing care.

**Keywords:** reason; unfinished nursing care; compromised care; integrative review

## 1. Introduction

In recent years, the phenomenon of missed nursing care (MNC), defined as care required by patients that nurses have planned and for various reasons delay or omit partially or completely, has been termed as unfinished nursing care (UNC) [1]. The latter has been established as an umbrella concept and includes all terminologies, theories and traditions developed in the field of MNC. Moreover, UNC has been recognised as an issue relevant to public health because of its potential consequences for patients, professionals, and healthcare organisations. It has also been emphasised that the occurrence of the UNC phenomenon affects citizens' trust in the National Health Service (NHS).

The lack of resources as the main reason, and the deterioration of service quality as the outcome, constitute the well-established evidence available on UNC over the years. However, the need for more efforts in this field of research to discover the reasons that promote and/or hinder the occurrence of UNC has been established; in fact, an in-depth understanding of the reasons can inform interventions to mitigate/prevent the phenomenon and avoid possible negative events [1].

Theoretically, it has been documented that UNC is influenced by factors on multiple levels, where the higher levels (e.g., policies regarding the amount of nursing care in units) can influence the lower levels and, ultimately, nurses' decision not to fulfil a patient's need [2,3]. Empirically, these assumptions have been tested in primary studies (e.g., [4]) in a real-world context, with a view of informing actions and strategies preventing the occurrence of UNC. Specifically, a recent systematic literature review summarised all primary studies published in the pre-pandemic era documenting the reasons for UNC [5]. The findings showed that factors at the unit (e.g., the resources available), nurse (e.g., priority setting abilities) and patient (e.g., the increased complexity of needs) level all play an important role in increasing the occurrence of UNC.

The body of evidence available has been further accumulated during the pandemic era, when studies conducted have revealed some changes in the factors triggering UNC; however, these studies [6–12] used mainly tools validated before the pandemic with the aim of comparing changes, if any, in the weight of different causes already known. Specific studies not using available tools, aimed at discovering new additional (and unknown) factors that may have played a role in triggering UNC during the pandemic, have not been summarised to date. Providing a summary of the empirical knowledge discovered in the field of reasons for UNC during the pandemic may: (a) describe changes in the causes of UNC in times characterised by unprecedented levels of pressures applied to the NHS; (b) inform new UNC mitigation and/or prevention interventions that may also be important in the post-pandemic era considering its long-term consequences; and (c) decrypt which factors most expose systems to unfinished care in pandemic times to inform future pandemic plans. Moreover, given the dramatic changes that incurred in the NHS due to the recent coronavirus (COVID-19) pandemic, re-evaluating the reasons for UNC can help the system, the executives, and the clinical nurses to make better decisions and set new priorities in their education, and implement policies to promote quality of care [13]. The purpose of the study was to describe the reasons for UNC as documented during the COVID-19 pandemic.

## 2. Materials and Methods

An integrative review was conducted following the Whittemore and Knafl's [14] methodological model, as it includes research from experimental and non-experimental studies to (a) extract results in a meaningful and systematic manner and (b) integrate evidence from various sources. This framework consisted of five steps: problem identification; literature search; data evaluation; data analysis; and presentation [14]. The Preferred Reporting Items for Systematic Reviews and Meta-Analyses (PRISMA) was followed for the identification, screening, eligibility, and inclusion stages of this review (Figure 1) [15].

### 2.1. Identifying the Research Questions

The review question was as follows: "what causes, factors, and predictors (here in after reasons) have been proven to trigger UNC during the pandemic?"

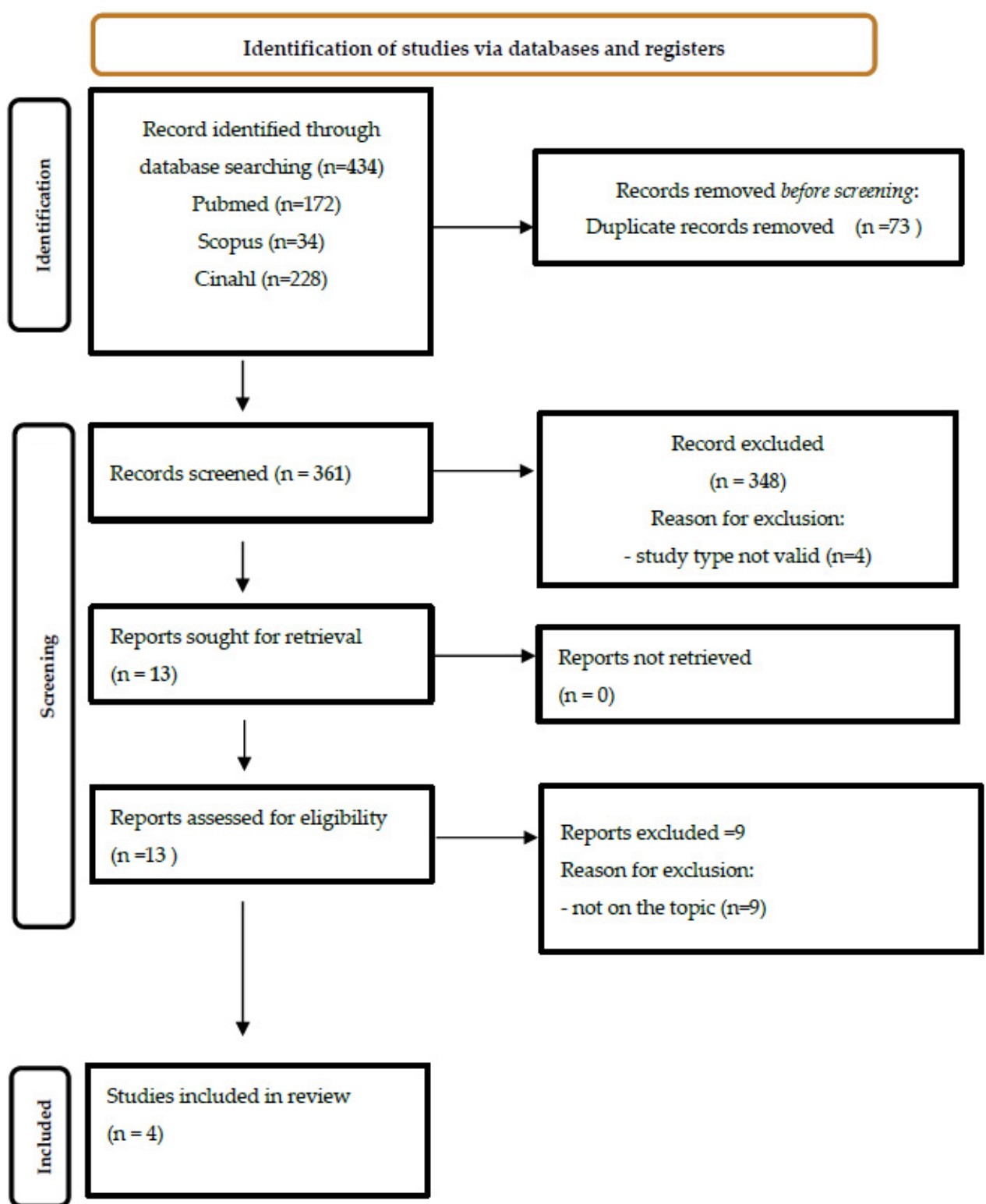

**Figure 1.** Flowchart of studies screening process (PRISMA guidelines) [16]. Abbreviations: PRISMA: Preferred Reporting Items for Systematic reviews and Meta-Analyses; CINAHL: Cumulative Index to Nursing and Allied Health Literature.

*2.2. Eligibility Criteria*

The literature search was conducted by consulting PubMed, Cumulative Index to Nursing and Allied Health Literature (CINAHL) and Scopus databases, using the following keywords: "missed nursing care", "rationing nursing care", "unfinished nursing care", "reasons", "causes" and "factors/predictors" (Supplementary Material Table S1). The following papers were included: (a) all primary studies relevant to the research question reporting (i) the abstract and (ii) the data collection period from 1 January 2020 to 1 May 2023 according to the official declaration of the starting and ending of the pandemic period [17]; (b) published in English, Italian or German; and (c) conducted with scientifically sound methodologies.

Studies using UNC measurement tools and assessing reasons according to these tools were excluded because they were developed and validated before the pandemic (e.g., MISSCARE Survey, Unfinished Nursing Care Survey), capturing factors established as relevant in that times. The pandemic has changed the organization and the process of health care systems and that of nursing care, thus the previous tools may not capture the realistic reasons triggering the UNC [6–10].

*2.3. Quality Appraisal*

A methodological quality assessment was carried out with the Critical Appraisal Skills Programme (CASP) for qualitative studies [18] and the Mixed Methods Appraisal Tool (MMAT) for mixed methods [19]. The grids required a judgement to be entered for each item (Y, Yes; N, No; CT, Can't tell) after having read each study carefully. The evaluation was conducted by two researchers (LS and SC, as well as other authors when the study was written by some of them) before they independently compared the findings. In the case of disagreements, a discussion meeting was held to reach a consensus. All identified studies showed sufficient quality, with 8/10 (CASP tool) and 13/17 (MMAT tool) (Supplementary Material Tables S2 and S3).

*2.4. Data Extraction and Synthesis*

On a preliminary basis, a data extraction grid was designed and piloted for clarity, feasibility, and utility in one study. No changes were required. Then, from the included studies, the following data were extracted and recorded in the grid: (a) authors, year of publication, country/study context, study period; (b) aim(s), type of study, data collection process; (c) sampling method, participants, demographic data (age, gender, professional experience); and (d) main results. The reasons for reported UNC were identified and extracted from each study; then, reasons extracted were categorised according to the levels where they originated (system, unit, nurse manager, nurse, patient) following the socioecological model as a reference (2). Subsequently, reasons were categorised and narratively described according to their similarities and differences.

## 3. Results

*3.1. Study Characteristics*

A total of 171 studies were identified and four were included [16] (Figure 1). These were all primary studies (Supplementary Material Table S4)—three based on a qualitative design [20–22] and one based on a cross-sectional design with an open-ended final question [23]. The studies were conducted in Italy [20,21], Finland [22] and Iran [23] in acute hospital settings [20–22]. The perspectives investigated were those of both healthcare professionals [20,22,23] and patients [21] involving from 14 [22] to 29 [20] participants in the qualitative studies and 462 [23] in the quantitative one. All studies were intended to explore the reasons for UNC as perceived during the pandemic, in the first months of 2021 [20,22,23] and between April and July 2022 [21], with well-designed and -conducted research methodologies.

*3.2. Reasons for UNC*

The UNC factors were categorised into system, unit, nurse manager, clinical nurse, and patient levels (Table 1).

**Table 1.** The reasons for unfinished nursing care in pandemics survey.

| Level | Theme | Subtheme | Authors (Year) |
|---|---|---|---|
| System | New healthcare system priorities | Dramatic changes due to the COVID-19 pandemic | Chiappinotto et al. (2023) [21] Safdari et al. (2023) [22] |
| | | Cost restraints | Chiappinotto et al. (2023) [21] |
| | Pre-existing frailty of healthcare facilities | Unsuitable environment layout | Chiappinotto et al. (2023) [21] Safdari et al. (2023) [22] |
| | | Old technologies | Chiappinotto et al. (2023) [21] Safdari et al. (2023) [22] |
| | | Discrepancies in resource allocation across units | Chiappinotto et al. (2023) [21] |
| | Poor support for nursing care | Lack of nurses and nursing care value | Chiappinotto & Palese (2022) [20] Safdari et al. (2023) [22] |
| | | System insensitive to UNC issues | Chiappinotto & Palese, (2022) [20] |
| | | High bureaucratisation and lack of investments in electronic records | Chiappinotto & Palese, (2022) [20] Hackman et al. (2023) [23] |
| | Challenges in leading nursing care | Lack of effective professional community | Hackman et al. (2023) [23] Safdari et al. (2023) [22] |
| | | High turnover | Hackman et al. (2023) [23] |
| Unit | Inappropriate care environment | Layout of the environment | Chiappinotto & Palese, (2022) [20] |
| | | High number of patients in each room | Chiappinotto & Palese, (2022) [20] |
| | | Chaotic environment | Chiappinotto & Palese, (2022) [20] |
| | Insufficient material resources | Material resources unavailable or limited | Chiappinotto & Palese, (2022) [20] |
| | | Restrictions in furniture/equipment | Safdari et al. (2023) [22] |
| | | Higher nurse/patient ratio | Chiappinotto & Palese, (2022) [20] Chiappinotto et al. (2023) [21] |
| | Insufficient human resources | Nurse shortages | Chiappinotto & Palese, (2022) [20] Chiappinotto et al. (2023) [21] Safdari et al. (2023) [22] Hackman et al. (2023) [23] |
| | | Nursing aide shortages | Chiappinotto & Palese, (2022) [20] |
| | | Physicians unavailable (e.g., off the unit) | Chiappinotto & Palese, (2022) [20] |
| | Ineffective inter- and intra-professional cooperation | Poor teamwork (lack of collaboration and communication/lack of in-group reflection on action) | Chiappinotto & Palese, (2022) [20] Safdari et al. (2023) [22] |
| | | Tension or communication breakdowns between nurses and medical staff, nurses and nursing aides, nurses and ward managers, and nurses and patients | Chiappinotto & Palese, (2022) [20] Safdari et al. (2023) [22] |
| | Ineffective shift design | Lack of staff during the day, nights, and weekends Excessive length of shifts | Chiappinotto & Palese, (2022) [20] Chiappinotto et al. (2023) [21] |

**Table 1.** *Cont.*

| Level | Theme | Subtheme | Authors (Year) |
|---|---|---|---|
| Unit | Ineffective unit organization and work process | Work process unpredictability due to unexpected internal (e.g., a new hospitalization, an urgency of a particular patient) or external (e.g., COVID-19) situations | Chiappinotto & Palese, (2022) [20] Hackman et al. (2023) [23] |
| | | Mission of the ward (specialised wards have a greater focus on the individual needs of patients) | Chiappinotto et al. (2023) [21] |
| | | Large number of discharges and admissions | Chiappinotto & Palese, (2022) [20] |
| | | Overlapping activities | Chiappinotto & Palese, (2022) [20] Chiappinotto et al. (2023) [21] Safdari et al. (2023) [22] |
| | | Limited capacity to react to unpredictable events (admissions/emergencies) | Chiappinotto et al. (2023) [21] |
| | | Ineffective routine | Chiappinotto & Palese, (2022) [20] Safdari et al. (2023) [22] |
| | | Lack of shared procedures | Chiappinotto & Palese, (2022) [20] Safdari et al. (2023) [22] |
| | | Higher frequency of interruptions | Chiappinotto & Palese, (2022) [20] Chiappinotto et al. (2023) [21] |
| | Ineffective models of nursing care delivery | Poor nursing care models of care delivery: functional nursing | Chiappinotto & Palese, (2022) [20] Chiappinotto et al. (2023) [21] |
| | | Incomplete or ineffective handovers | Chiappinotto & Palese, (2022) [20] |
| Manager | Inadequate nurse manager's leadership | Inadequate nurse manager's leadership | Chiappinotto & Palese, (2022) [20] Chiappinotto et al. (2023) [21] |
| Nurse | Nurses' attitudes, competences and performances | Being in a hurry | Chiappinotto et al. (2023) [21] |
| | | Reduced work capacity due to increased age | Chiappinotto & Palese, (2022) [20] |
| | | Lack of experience, knowledge, competences (e.g., empathic) | Chiappinotto & Palese, (2022) [20] |
| | | Lack of responsibility | Chiappinotto & Palese, (2022) [20] Chiappinotto et al. (2023) [21] |
| | | Low motivation | Chiappinotto & Palese, (2022) [20] Chiappinotto et al. (2023) [21] Hackman et al. (2023) [23] |
| | | Higher stress, fatigue | Chiappinotto & Palese, (2022) [20] Chiappinotto et al. (2023) [21] Hackman et al. (2023) [23] |
| | | Poor time management skills | Chiappinotto & Palese, (2022) [20] |
| | | Ineffective delegation skills | Chiappinotto & Palese, (2022) [20] Chiappinotto et al. (2023) [21] |
| | | Ineffective priority-setting skills | Chiappinotto & Palese, (2022) [20] |
| | | Wrong nursing care planning | Chiappinotto & Palese, (2022) [20] Safdari et al. (2023) [22] |

**Table 1.** *Cont.*

| Level | Theme | Subtheme | Authors (Year) |
|---|---|---|---|
| Nurse | Weaknesses in education | Incomplete training/mentoring (in the transition as a newly qualified graduate)/inadequate orientation of the new staff | Chiappinotto & Palese, (2022) [20] Hackman et al. (2023) [23] |
| | | High turnover among nurses | Chiappinotto & Palese, (2022) [20] |
| | Poor humanistic view of patient care | Nursing care not patient-centred | Chiappinotto & Palese, (2022) [20] |
| Patient | | Clinical instability | Chiappinotto & Palese, (2022) [20] Hackman et al. (2023) [23] |
| | Increased demand of patient care | Complexity/needs Worse clinical conditions | Chiappinotto & Palese, (2022) [20] Hackman et al. (2023) [23] Safdari et al. (2023) [22] |
| | | Age | Safdari et al. (2023) |
| | | Cognitive impairments | Chiappinotto & Palese, (2022) [20] Safdari et al. (2023) [22] |
| | | Loneliness | Chiappinotto & Palese, (2022) [20] |
| | Lack of carer support | The absence of relatives/caregivers Hospital restriction to relatives | Chiappinotto & Palese, (2022) [20] Safdari et al. (2023) [22] |
| | Increased nursing care needs and care expectations | Demanding patients | Chiappinotto & Palese, (2022) [20] Chiappinotto et al. (2023) [21] |

Abbreviations: ADL, activities of daily living; UNC, unfinished nursing care.

### 3.2.1. System Level

The system level is defined as the highest organisational level that influences policies, programmes, and culture of the entire institution, and is capable of triggering UNC [2]. At this level, available studies have identified the following reasons for UNC:

- New priorities of the health system. The healthcare system has undergone major reorganisation, related to the drastic changes due to the COVID-19 pandemic, which has further reduced resources by exacerbating the pre-existing issues of the system [20,22] and leading to cost restraints in some sectors to rendering available resources to others [21].

- Pre-existing frailty of healthcare structures and processes. The inadequate environments, such as the old layouts of hospital buildings [21,22], as well as the discrepancies in the allocation of resources across units, have been seen as pre-existing frailties that have been exacerbated during the pandemic, thus increasing the risk of UNC. The structural and processual factors combined with an unbalanced workforce across units, and poor environments (e.g., distance between units), have been reported as leading to UNC [21].

- The poor support for nursing care. Systems causing a lack of nurses at the unit level [20,22], and not emphasising and/or communicating internally and externally the role and the value of nurses, have been documented as increasing the risk of UNC. Moreover, those systems not considering appropriately the early signs of poor-quality care (e.g., by analysing incident reports) have been perceived as insensitive towards UNC issues, neglecting its relevance and consequently strategies aimed at preventing it. In addition, some systems perpetuated some UNC factors when they did not invest in technologies facilitating nursing care [20]: the high level of bureaucratisation increased further during the pandemic (e.g., the need to collect certifications and to

check issues) and led to time being wasted on administrative tasks leading nurses to postpone care [20,23].

- Increased challenges in leading nursing care. The fragmentation in the community of nurses as a profession and as a system has been reported as affecting its capacity to effectively address changes and policies, as an independent body, both at the political and institutional levels [22,23]. Similarly, the increased nursing turnover [23] has been reported as triggering UNC.

3.2.2. Unit Level

The unit level, as that lived by both nurses and patients, reflects the context where UNC occurs, and where some additional factors have played a role during the pandemic:

- Inadequate care environment. The environments within the units have been reported as inadequate in terms of their layouts, leading to time being wasted by nurses in getting to rooms or retrieving material. Moreover, with many patients being in the same room, the increased attention and processes needed to protect them from safety issues required more time and, when nurses experience a lack in resources, a greater occurrence of UNC. In some units, the perceived chaos and confusion distracted nurses while they were providing the necessary care [20].
- Insufficient material resources. Material resources [20], both in terms of supplies and equipment [22], were poorly available or limited: nurses have been reported as spending time searching for them in other units, postponing the care required [20,22].
- Insufficient human resources. The lack of human resources, reflected in the high nurse/patient ratio [20,21], due to the shortage of nurses [20–23] and nursing aides [20] has been documented. In addition, the absence of physicians (e.g., when they are outside the unit) also increased workloads, resulting in some care needs being missed [20].
- Ineffective shift design. An adequate presence of staff was not always planned during the day, at night and at weekends; the length of shifts was also a problem with shifts being too long. The idea that there are fewer care activities to provide at the weekend generated ineffective shift planning, reducing the number of nurses at the unit level during weekends [20,21]; on the other hand, excessively long shifts led to fatigue and lowered the standard of nursing work.
- Ineffective unit organization and work process. The mission of the unit and its continuing change have triggered uncertainty regarding the priorities [21]. Specifically, nurses were continuously called to review their work processes, redefining priorities and activities [20], due to continuous unexpected events [20] related to internal (e.g., emergencies) and/or external (e.g., COVID-19 patients) new conditions [23] such as the high number of discharges and admissions [20]. The continuous need to redefine care plans was also influenced by the high frequency of interruptions (e.g., answering the telephone) [20,21] and the disrupted routines due to changes imposed on the work processes in managing the pandemic [20,22]. The several competitive activities [20–22] have increased the occurrence of UNC. The high number of newly qualified nurses, deployed from other wards, prevented the possibility of working with shared procedures [20,22], leading to uncoordinated decisions, the wasting of time and ultimately UNC [20,21]. Expanding the capacity of the unit in response to the numerous unpredictable events was not always possible; therefore, with the same resources, all patients were cared for, but not all care needs were catered for, thereby increasing the occurrence of UNC.
- Ineffective models of nursing care. The delivery models did not support the personalisation of care expected both by patients and nurses. Specifically, the functional model in which tasks are fragmented, accompanied by the need to limit the exposure to patients due to the risk of contagion, have been reported as threatening care needs; also, handovers were incomplete or ineffective, due to the fragmentation of care, with failure to communicate patient needs ultimately leading to UNC [20].

- Ineffective inter- and intra-professional collaboration. The lack of collaboration and communication inside the nursing profession and across various professionals has been reported as causing tensions or interruptions in communication during the care process, thus increasing the risk of UNC [20,22].

### 3.2.3. Nurse Manager Level

At the nurse manager level, inadequate leadership, lacking in clear and shared aims and interest in the professional protection and growth of the nurses in difficult times has been reported to increase the occurrence of UNC [20,21].

### 3.2.4. Clinical Nurse Level

The issues belonging to the clinical nurse level are those strictly related to each individual nurse and may all contribute to UNC.

- Issues with nurses' attitudes, competences and performances. The lack of empathy triggered poor communication and understanding of needs, and while working in a hurry prevented any contact with patients, thereby threatening the capacity to identify patients' needs [21]. Moreover, reduced working abilities related to an increase in age [20], and a lack of work experience, knowledge and skills [20], as well as professional responsibility [20,21] and/or motivation [20,21,23], have also been reported as increasing the occurrence of UNC. Furthermore, the tiredness caused by high workloads [20,21,23] and the poor ability to manage time, to attribute priorities [20] or to delegate [20,21] have generated UNC. Errors in care planning (for example, scheduling of unnecessary interventions) have also been underlined as leading to UNC [20,22].
- Weaknesses in education: incomplete training or mentoring [20,23] led to long periods of time being needed to work effectively as an independent nurse among those just introduced into the unit. An increased risk of missing under-recognised needs was also reported. On the other hand, excessive burden among some more experienced nurses has been documented as causing a high nursing turnover, which implied the need to support new colleagues [20].
- Poor humanistic vision of the patient. Nursing care not centred on the person, but rather on the activities/tasks to be provided, forced by the extreme working conditions experienced, have reduced the capacity to consider all needs (for example emotional ones) that have been missed [20].

### 3.2.5. Patient Level

The last level identified, related to patients, underlines the important change in the patient profile.

- Increased demand for patient care. During the pandemic, an increased number of patients were in unstable conditions [20,23], with highly complex and/or worse clinical conditions [20], many with co-morbidities [22,23], and elderly people with cognitive decline [20,22] and living alone [20]. These patients required more care, as they were not always able to communicate their needs, and above all, they were not supported by caregivers [20,22,23].
- Lack of carer support. During the pandemic, relatives could not access the hospitals due to the restrictive policies; consequently, the simplest care activities [20,22] often delegated to families were not performed.
- Increased nursing care needs and expectations. In some contexts, patients became more demanding; they also resisted treatments because they did not believe that the pandemic and the need for treatment were truthful; for example, they rejected educational interventions regarding vaccinations [20,21].

## 4. Discussion

Only four studies have investigated the reasons for UNC during the pandemic without using tools using a predefined set of UNC causes: on the one hand, using predefined tools as many researchers did [24] may provide valid and comparable evidence, whereas on the other hand, innovative approaches may provide new insights on additional factors influencing the occurrence of UNC during challenging times like those lived through in the pandemic. Qualitative studies were mainly performed during the pandemic [20–22], providing innovative perspectives from those who were experiencing the issue. Nurses' perceptions have mostly been investigated, at the bedside and at the different levels of healthcare services [20,22,23]. It has been widely recognised that the nursing workforce has been affected by the pandemic [25,26]; therefore, giving them a voice by involving all levels from the bedside to the executive one is important. However, the patients' perspective has been investigated in only one study, so in the pre-pandemic era their perspectives remained mostly neglected. The patients' perceptions are important [20] in valuing their reported experience (e.g., Patient-Reported Experience Measures (PREMs)) given that unfinished care is mostly related to their expectations.

At the overall level, all participants were expert informants according to their professional experience, age or experience with hospital care. Therefore, the reasons for UNC that emerged reflect those lived by experts that may have compared the pre-pandemic professional experience with that encountered during the pandemic. However, two studies have been conducted in Italy [20,21] reflecting on the country most affected by the pandemic, forcing the adoption of urgent—and unprecedented—healthcare policies that made significant changes to the nursing care; others were conducted in Iran and Finland. Therefore, the reasons for UNC reflect specific contexts, and more research is needed in the future to accumulate more evidence, but it must be conducted with good-quality methodologies despite the difficult times experienced also affecting hugely the research capacity.

*The Reasons for UNC*

To date, reasons triggering UNC have been documented by measuring their significance over a list of potential causes listed in the MISSCARE survey (e.g., [6–9]) and in the Unfinished Nursing Care Survey (e.g., [10]). In this context, the lack of staff (e.g., the inadequate number of nurses) [7–9], or the increased number of patients [7,10], as well as their unpredictable clinical condition [7,10], or some issues in making the right priorities [10], have been established as facilitating UNC. Specifically, factors were identified in the MISS-CARE survey [27], namely communication, labour resources and material resources, and UNC [3] factors have been identified in terms of communication, prioritisation, supervision of nursing assistants, material resources, human resources, and predictability of workflow.

During the COVID-19 pandemic, some additional reasons emerged at the system, unit, nursing management, clinical nurse and patient levels. At the theoretical level, Jones had already established the importance of some factors above the simple unit that were capable of applying negative forces leading to UNC [2]. These factors, set at the system level, suggest that unfinished care is not an isolated phenomenon but mirrors the values, priorities, investments and strategic plans of the entire system towards nursing care. During the pandemic, at this level it also emerged that the contribution of the nurses as a profession or body has been perceived as important in representing, claiming and addressing the policies. Therefore, the empirical studies performed during the pandemic confirm the theoretical framework of Jones regarding the importance of the system by adding the role of the professional bodies; however, all these elements should now be operationalised and measured to weigh their contribution, compare their relevance in the context of other factors at the micro level and to benchmark across countries [2]. In the traditional approach of UNC studies, bedside nurses have been involved in ranking the causes at the unit level; in different systems, the same reasons emerged with slight differences during the pandemic [28]. Possibly, some factors at the system level may modulate the occurrence by applying negative or positive forces that merit being discovered.

At the unit level, which was mostly investigated in pre-pandemic studies with tools (29), new reasons appear linked to ineffective work and organizational processes [20,22] and to the design of shift work, which also considers the use of personal protective equipment (PPE) [20,21]. The units were exposed to major revisions, of limited duration. Previous routines were destroyed, and the new nurses hired could not always be helped to aid their understanding of the work. The continuous internal and external unforeseen events have further weakened the organizational structure and work processes; furthermore, the modification of the patients' needs [28,29] has created new priorities that have probably not been understood. Some reasons (for example, problems related to shifts) may be addressed with already established evidence [5], and others (for example, problems related to ward organisation and models of nursing care delivery) with disaster management strategies. Above all, at the unit level, a new reason for UNC emerged concerning the leadership of the nurse manager [20,21]: being close to the nurses, guiding and supporting them in the extreme conditions experienced, is challenging. Therefore, preparing future leaders for facing prolonged challenging situations might be important.

Factors related to nurses have also been identified previously: in the pre-pandemic era, the reasons focused more on the experience of nurses, on the mix of skills [5], while in the pandemic era the educational gap is more evident [20,22], influencing competences, skills and also attitudes that may impact negatively on patients (errors, infections and low satisfaction with care) [30]. These new reasons for UNC coincide with the main challenges that nurses have faced in this period in dealing with the emergency and managing changes. The high intentional turnover (moving nurses from one department to another in urgent situations) has made it difficult to ensure the appropriate training; on the other hand, limiting the clinical rotations [31] due to the pandemic may have prevented the development of competence during nursing education. Universities should refocus their education and priorities, and hospitals must identify adequate introductory plans, designing one for routine times and a second one for dealing with crises/disasters.

Finally, during the pandemic the care demand has increased significantly in all systems: therefore, it is not surprising that UNC was also triggered by the patients' condition. In many systems, relatives were involved in contributing to nursing care by compensating for the nursing shortage; the restrictions also imposed on volunteers have made the need for nurses even more evident. The increased needs of patients and the unavailable nursing care have generated moral distress [32]; the same values and beliefs of patients (for example against vaccinations, refusing treatments because the pandemic 'does not exist'), in contrast to what was happening, made the relationship difficult, creating tensions and difficulties in ensuring the care needed was delivered.

The map of factors that emerged can help identify other strategies to be included in future pandemic plans in an interdisciplinary approach [33]. Nurse executives and managers are crucial in creating positive professional environments aimed at supporting professionals and work processes, through organisational models of care that ensure the support of professionals in decision-making, good practice, and patient safety [34]. Nurse educators can promote awareness among new professionals. Researchers can design studies facilitating the implementation of the discovered strategies in different contexts.

We conducted an integrative review; however, the language limitations and the publication time lag may have introduced some selection bias. Studies conducted during the pandemic were included, and others may be in the process of being published. Therefore, updating this review is recommended. Moreover, some studies have investigated reasons with different methodologies, sometimes as predictors/factors and others as experiences. We used these concepts interchangeably, even if they have different meanings, as reasons associated with the UNC phenomenon/factors as influencing the occurrence of UNC. In the future, it will be necessary to differentiate their contribution by summarising the evidence produced in each (Supplementary Material Table S5).

## 5. Conclusions

To the best of our knowledge, this is the first integrative review summarising the reasons for UNC as reported in primary studies during the pandemic. Taking the socio-ecological model as a reference, the reasons that emerged affected five levels, namely the system, the unit, the nurse manager, the clinical nurse and the patient. New reasons emerged as compared to the pre-pandemic literature suggesting that the UNC is also triggered by some pre-existing frailties of the NHS regarding nursing care. The map of reasons that emerged may be used in informing future pandemic plans as a complex intertwined and multilevel phenomenon that suggests a need for a systemic approach.

**Supplementary Materials:** The following supporting information can be downloaded at: https://www.mdpi.com/article/10.3390/nursrep14020058/s1, Table S1: Search strategies used in approached databases; Table S2: Study quality Assessment: Critical Appraisal Skills Programme (CASP) for a Qualitative Research (Critical Appraisal Skills Programme [18]); Table S3: Study quality Assessment: Mixed-Method Appraisal Tool (MMAT) [19]; Table S4: Description of included studies; Table S5: Study Limitations.

**Author Contributions:** Conceptualisation, L.S., S.C., R.M., P.R. and A.P.; methodology, S.C.; software, R.M. and P.R.; validation, L.S. and S.C.; formal analysis, L.S. and S.C.; investigation, L.S. and S.C.; resources, S.C.; data curation, A.P.; writing—original draft preparation, L.S. and S.C.; writing—review and editing, R.M., P.R. and A.P.; visualization, S.C.; supervision, R.M., P.R. and A.P.; project administration, A.P. All authors have read and agreed to the published version of the manuscript.

**Funding:** This research received no external funding.

**Institutional Review Board Statement:** Not applicable.

**Informed Consent Statement:** Patient consent was not required as this is a review of existing published studies.

**Data Availability Statement:** All data connected to this review are available within the online Supplementary Materials sections.

**Public Involvement Statement:** There was no public involvement in any aspect of this research.

**Guidelines and Standards Statement:** This manuscript was drafted against the Preferred Reporting Items for Systematic Reviews and Meta-Analysis (PRISMA) [16].

**Use of Artificial Intelligence:** AI or AI-assisted tools were not used in drafting any aspect of this manuscript.

**Conflicts of Interest:** The authors declare no conflicts of interest.

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
