# Peer review of "The Reasons for Unfinished Nursing Care during the COVID-19 Pandemic: An Integrative Review"

_nursrep, doi:10.3390/nursrep14020058_

Round 1
Reviewer 1 Report
Comments and Suggestions for Authors
Dear editor and dear authors,
Thank you for the opportunity to review your paper entitled “The reasons of Unfinished Nursing Care during the COVID-19 pandemic: an integrative review. I appreciated the paper; here are some suggestions.
The introduction, methodology, results, and discussion sections are well-framed.
In this study, the authors explore factors affecting unfinished nursing care during a pandemic. The introduction is well-written, and the authors justified the topic well.
This is a significant theme. During the manuscript, the author explained which factors change with a pandemic, and they made a good comparison.
The materials and methods, the search strategy, the clinical appraisal and the description of studies are in Supplementary Material S5; I suggest giving this information to the readers.
In the result of Figure 1 - PRISMA, the subtitle needs to be corrected (lines 129 and 130). The authors did not search PsycINFO.
The discussion is well-designed. Regarding limitations, the authors sent us the file in Supplementary Materials S5. They did a good analysis of the limitations, so I suggest presenting the main topics in the article.
The conclusions are consistent with the evidence and arguments presented. We suggest explaining how we can translate the evidence from this research to clinical practice. The references are appropriate.
Thank You
Author Response
Dear Editor
We really thank the reviewers for their valuable comment on our manuscript entitled "The reasons of Unfinished Nursing Care during the COVID-19 pandemic: an integrative review".
The criticisms and suggestions provided have been considered with care as a great opportunity to improve the manuscript.
We undertook immediately the revision in order to be timely. As suggested, each changed has been highlighted in red colour. An additional editing proof has been also undertaken.
Overall, we performed the following changes:
- The abstract has also been revised and arranged with abbreviations.
- The methods have been revised to improve clarity: in addition to explaining the methodology used, the selection criteria have been clarified and Figure 1 has been fixed.
- The discussions were reviewed highlighting how the results can have an impact on clinical practice. In addition, the limits have been added in a discursive manner.
Thank you again for the opportunity to revise the manuscript.
On behalf of the authors
COMMENTS FOR THE AUTHOR:
Reviewer #1:
Dear Reviewer,
Thank you very much for your consideration of our research, which we have addressed as follows, highlighting the changes in red.
- Dear Editor and Authors,
Thank you for the opportunity to review your paper entitled “The reasons of Unfinished Nursing Care during the COVID-19 pandemic: an integrative review. I appreciated the paper; here are some suggestions.
The introduction, methodology, results, and discussion sections are well-framed.
In this study, the authors explore factors affecting unfinished nursing care during a pandemic. The introduction is well-written, and the authors justified the topic well.
This is a significant theme. During the manuscript, the author explained which factors change with a pandemic, and they made a good comparison.
Thank you very much for your consideration of the research.
- The materials and methods, the search strategy, the clinical appraisal and the description of studies are in Supplementary Material S5; I suggest giving this information to the readers.
Thank you very much for this suggestion, we have included this information to the readers in the text.
-In the result of Figure 1 - PRISMA, the subtitle needs to be corrected (lines 129 and 130). The authors did not search PsycINFO.
Thanks for the feedback, the title has been corrected and so have the abbreviations.
-We suggest explaining how we can translate the evidence from this research to clinical practice.
Thanks for the suggestion, we have added evidence from this research to clinical practice in the discussion.
- Regarding limitations, the authors sent us the file in Supplementary Materials S5. They did a good analysis of the limitations, so I suggest presenting the main topics in the article.
Thank you very much for this important suggestion, we have included the limitations in the discussion in a discursive way.
- The conclusions are consistent with the evidence and arguments presented. We suggest explaining how we can translate the evidence from this research to clinical practice. The references are appropriate.
Thank you for the suggestion. We have added a section regarding this issue at the end of the discussion.
Reviewer 2 Report
Comments and Suggestions for Authors
Overall the manuscript is well-written and focuses the major points/issues of UNC, during pandemic.
Please take attention to abbreviations on Abstract.
On Methods section you should specify why not using UNC tools. Do you believe those tools do not measure all UNC, specifically during COVID-19?
However, you present an integrative review, and your question directs to a Scoping Review, where Articles Quality Appraisal is not done. Why did you choose an integrative? Where there steps you skipped from Systematic Reviews?
Kind regards,
Comments on the Quality of English Language
Review minor english language mistakes and abbreviations order.
Author Response
Reviewer #2:
Dear reviewer,
Thank you for your precise and timely suggestions, which we have addressed below by highlighting the changes in red.
- Overall the manuscript is well-written and focuses the major points/issues of UNC, during pandemic. Take attention to abbreviations on Abstract.
Thanks to your feedback, we have revised not only the abstract but also the entire manuscript.
-On Methods section you should specify why not using UNC tools. Do you believe those tools do not measure all UNC, specifically during COVID-19?
Thank you for this point - we have incorporated the comment to make the selection criteria clearer.
-However, you present an integrative review, and your question directs to a Scoping Review, where Articles Quality Appraisal is not done. Why did you choose an integrative? Where there steps you skipped from Systematic Reviews?
Thank you for your suggestion. We have explained the methodology used with all the stages carried out.
Yours sincerely
Stefania Chiappinotto on behalf of the authors
Reviewer 3 Report
Comments and Suggestions for Authors
Introduction
- To raise the study urgency, please highlight the specific problem “What is the impact of UNC during COVID-19?”. Does the UNC generate a new significant problem for patients, hospitals, or other related parties, such as economic views? So, this study can reveal why this research topic needs to be conducted.
Method
- I recommend to use the Cochrane risk of bias tool to asses the study quality
Discussion:
- Since the study result revealed some main points of reason, I suggest discussing them one by one to make it clear.
- On page 11, lines 360-365, the author presented the study suggestion. However, this point needs to be more detailed and more technical to make it clearer.
Author Response
Dear Editor
We really thank the reviewers for their valuable comment on our manuscript entitled "The reasons of Unfinished Nursing Care during the COVID-19 pandemic: an integrative review".
The criticisms and suggestions provided have been considered with care as a great opportunity to improve the manuscript.
We undertook immediately the revision in order to be timely. As suggested, each changed has been highlighted in red colour. An additional editing proof has been also undertaken.
Overall, we have made the following changes:
(a) Introduction was revised and arranged to emphasise importance of the topic
(b) Discussions have been revised to emphasise how results can have an impact on research.
Thank you again for the opportunity to revise the manuscript.
On behalf of the authors
Reviewer #3:
Dear reviewer,
Thank you for your precise and timely suggestions, which we have addressed below by highlighting the changes in red.
Introduction
- To raise the study urgency, please highlight the specific problem “What is the impact of UNC during COVID-19?”. Does the UNC generate a new significant problem for patients, hospitals, or other related parties, such as economic views? So, this study can reveal why this research topic needs to be conducted.
Thanks to your feedback, we have revised the introduction emphasising these aspects
Methods
- I recommend to use the Cochrane risk of bias tool to asses the study quality
Thanks for the suggestions. However, we decided in the review protocol choice to use two different tools, the Critical Appraisal Skills Programme (CASP) for qualitative studies and the Mixed Methods Appraisal Tool (MMAT) for mixed methods because they are considered by the bibliography to be reliable [1] and with specific criteria for assessing the methodological quality of the studies [2].
- Pace, R., Pluye, P., Bartlett, G., Macaulay, A. C., Salsberg, J., Jagosh, J. & Seller, R. (2011) Testing the reliability and efficiency of the pilot Mixed Methods Appraisal Tool (MMAT) for systematic mixed studies review. International Journal of Nursing Studies, 49 (1), 47–53.
2 Ma, L. L., Wang, Y. Y., Yang, Z. H., Huang, D., Weng, H., & Zeng, X. T. (2020). Methodological quality (risk of bias) assessment tools for primary and secondary medical studies: what are they and which is better?. Military Medical Research, 7(1), 7. https://doi-org.ezproxy.unibo.it/10.1186/s40779-020-00238-8
- Discussion:
- Since the study result revealed some main points of reason, I suggest discussing them one by one to make it clear.
- On page 11, lines 360-365, the author presented the study suggestion. However, this point needs to be more detailed and more technical to make it clearer.
Thank you for this observation - we have arranged the discussions to make them clearer.
Yours sincerely
Stefania Chiappinotto on behalf of the authors